# Scoping Review on Interventions for Physical Activity and Physical Literacy Components in Brazilian School-Aged Children and Adolescents

**DOI:** 10.3390/ijerph18168349

**Published:** 2021-08-06

**Authors:** Valter Cordeiro Barbosa Filho, Wallingson Michael Gonçalves Pereira, Bianca de Oliveira Farias, Thereza Maria Magalhães Moreira, Paulo Henrique Guerra, Ana Carolina Melo Queiroz, Victor Hugo Santos de Castro, Kelly Samara Silva

**Affiliations:** 1Post-Graduate Program in Collective Health, Ceara State University, Fortaleza 60714-903, Brazil; wallingsonmichaelgp@hotmail.com (W.M.G.P.); biancafariasnutri@gmail.com (B.d.O.F.); tmmmoreira@gmail.com (T.M.M.M.); acmq28@hotmail.com (A.C.M.Q.); vsantosdecastro@gmail.com (V.H.S.d.C.); 2Federal Institute of Education, Science and Technology of Ceara, Aracati 62930-000, Brazil; 3Federal University of Fronteira Sul, Chapeco 89802-112, Brazil; paulo.guerra@uffs.edu.br; 4Research Center for Physical Activity and Health, Federal University of Santa Catarina, Florianopolis 88040-900, Brazil; ksilvajp@gmail.com

**Keywords:** physical literacy, physical activity, intervention, children, adolescents

## Abstract

This scoping review mapped the existing evidence on interventions to promote physical activity (PA) and/or components of physical literacy (PL) in Brazilian school-aged children and adolescents. Nine electronic databases and gray literature were consulted in May 2020, with no limit on year or language. School-based intervention studies (6 to 18 years old, primarily) that assessed PA or PL components (PA-related factors or attributes) were eligible. The studies were stratified by children (<12 years of age) and adolescents (≥12 years of age). A total of 63 documents were included, which refer to 42 different intervention studies. Twenty-five interventions focused on adolescents and 17 on children. The most-used strategies in the interventions were changes in physical and environmental education classes, extracurricular PA sessions, and health education. No study has analyzed all components of PL or evaluated PL using specific protocols or instruments. PA attributes were the most studied components (30 studies). This review identified the need to conduct interventions with strategies that target all components of PL, representing important elements for a research agenda that underlies school interventions that contribute to an active lifestyle.

## 1. Introduction

The World Health Organization’s Global Action Plan on Physical Activity (PA) aims to increase PA and reduce sedentary behavior by guiding the creation of active societies, environments, people, and systems by 2030 [1]. Therefore, actions are proposed to transform social norms and attitudes that include enjoyable, affordable, socially, and culturally appropriate experiences of PA, which provide mass participation, behavior change, health, and physical literacy (PL). In addition, the document highlighted that a school-based environment reinforces lifelong health and PL according to individual, social, cultural, and economic features. This universal document invites the world to perceive PL as an important component of individual and collective actions to promote PA and health in the global population.

PL refers to a set of characteristics or attributes that expand an individual’s potential to become involved and maintain the practice of PA [2]. Thus, PL components involve not only the PA behavior (e.g., active commuting to school, regular PA practice in sports, and others) but also PA-related factors (e.g., motivation, self-efficacy, social support, and environmental factors) and attributes (e.g., physical fitness and motor behavior) [3,4,5] that incorporate experiences and enrich an active lifestyle. Although there is no direct empirical test of the effect of PL on health, evidence supports that some of the core components of PL, particularly motor competence, motivation, and affect, may influence PA behavior and, by extension, health outcomes [6].

Consequently, interventions aimed at promoting PA should assess beyond the behavior to recognize the benefits of these interventions in different aspects that involve movements (attributes and associated factors of PA). Some countries, such as Canada, the United Kingdom, and Australia, have advanced in studies on the evaluation and promotion strategies of PL in children and adolescents [2,7]. In particular, a systematic review was published in 2012 [8] and synthesized data from 129 intervention studies that examined PL’s effects on PA factors, behavior, and attributes. However, no intervention with children and adolescents from low- and middle-income countries has been reported [8]. Low- and middle-income countries have high rates of physical inactivity among children and adolescents, and they are at high risk for social and environmental elements (e.g., violence and unstructured schools) that may negatively impact PA and PL components [9]. Thus, improving PA, PL, and health in this context is part of the global health agenda [1].

In Brazil, a middle-income country, some reviews have summarized important evidence on the promising results of interventions to promote PA in children and adolescents [10,11,12]. However, these reviews have only presented information about PA behavior but have not considered the breadth and relevance of the different components of PL (such as motor behavior, physical fitness, and social and psychological aspects of PA). The different components of PL contribute systematically to the adoption and maintenance of an active lifestyle in children and adolescents and on the health of this population [6]. Therefore, a review that summarizes the methodological aspects, intervention strategies, and main results on this theme can guide future studies and practices on interventions to promote the PA, PL, and comprehensive health of school-aged children and adolescents.

Thus, the present scoping review aimed to map intervention studies to promote PA or other PL components among Brazilian school-aged children and adolescents. Our review is based on the following question: What is the existing evidence on interventions aimed at promoting PA or other PL components—PA factors and attributes—in Brazilian school-aged children and adolescents?

## 2. Materials and Methods

### 2.1. Protocol

This scoping review followed the recommendations of the Joanna Briggs Institute manual [13] and was reported in accordance with the guidelines recommended by the Preferred Reporting Items for Systematic Review and Meta-Analyses Extension for Scoping Reviews (PRISMA-ScR) [14], as detailed in Appendix A. However, this study was not registered.

### 2.2. Eligibility Criteria

Considering the review question, the eligibility criteria were based on the population concept–context domains.

(I) Concept: Studies were included if they performed any intervention strategy that aimed at promoting PA or any components of PL. The components of PL were operationalized in modifiable PA-related factors (e.g., motivation, self-efficacy, interpersonal support, and environmental factors), PA behavior (e.g., weekly volume of PA), and attributes (e.g., physical fitness and motor behavior) [3,6]. PA was considered based on the WHO global action plan to promote PA [1], which is understood as any movement performed of a musculoskeletal nature that has energy expenditure as a result, and PA measurements were included regardless of the type or domain of PA (leisure, work, and active commuting) or type of measurement (subjective and objective). Modifiable PA factors were considered based on the following dimensions proposed by Keegan et al. [15]: cognitive, social, psychological, and physical dimensions. They were included independently on the scales and instruments used to estimate modifiable PA factors. PA attributes include cardiorespiratory and musculoskeletal fitness, flexibility, balance, coordination, and body composition (e.g., body mass index (BMI)) as its components [5,6], and these variables can be measured using direct or field tests.

(II) Population and Context: Studies were included if they evaluated children and adolescents (aged 6 to 18 years) enrolled in Brazilian schools. Studies that exclusively considered specific populations (e.g., children with musculoskeletal, neurofunctional, or metabolic diseases) were not included.

No restrictions were considered for experimental designs (all experimental studies), including studies without randomization between groups and no control groups. No limits were imposed on the publication year or language of the articles.

### 2.3. Sources of Evidence

The search was performed in May 2020 and updated in June 2021 (Appendix A) in the following electronic databases: ERIC, Embase, MEDLINE/PubMed, LILACS, PsycINFO, SciELO, Scopus, SPORTDiscus, and Web of Science. Complementary searches were performed in the Brazilian Digital Library of Theses and Dissertations (BDTD), Google Scholar, specialized websites (Brazilian Society on Physical Activity and Health or SBAFS and *Projeto Esporte Brasil* or PROESP-BR), and was supplemented by a manual search of references in the reference lists of articles, aiming at finding relevant studies that were not included in the electronic databases (including gray literature).

### 2.4. Search Strategies

The search strategies included terms obtained from the existing literature in the area and were refined through consultations with specialists in the field. The search was structured in both Medical Subject Headings (MeSH) and text words presented in the literature. The terms were combined using truncation symbols and Boolean operators (“OR”, “AND”, and “NOT”). The search equations are presented in Appendix A.

### 2.5. Study Selection

The studies were incorporated into a library in EndNote Web after the search, and duplicates were eliminated before the selection process. At least two reviewers (W.M.G.P., V.H.S.d.C. and/or B.d.O.F.) independently performed the selection process, and the discrepancies between the reviewers were resolved by a third reviewer (V.C.B.F.). The study selection was performed in two stages: (1) screening of titles and abstracts and (2) reading of the full text. Only studies that met all eligibility criteria were included in the final report. All studies excluded at this stage were duly justified in the flow chart below according to the recommendations of PRISMA-ScR [14].

### 2.6. Data Extraction

One of the three reviewers conducted this phase (W.M.G.P., B.d.O.F. and A.C.M.Q.) using a standardized data extraction spreadsheet and the extracted data were revised by another reviewer. The extracted information included the characteristics of the studies (reference, year, place and type of study, and region in Brazil), population (sample size, school year, number of schools, age group, and eligibility criteria), interventions (type of intervention, time, and intervention strategies), and outcomes (measurement and results on PA factors, PA behavior, and/or PA attributes). Considering that this was a scoping review and aimed to map research independently of its quality, the risk of bias was not assessed [13].

### 2.7. Data Synthesis

A descriptive synthesis of the studies was adopted to describe the characteristics of the publications retrieved in this review (population, context, and concept). The components of PL that were evaluated in the included studies were presented in three domains: PA-related factors (psychological, social-environmental, and cognitive, such as self-esteem, satisfaction, self-efficacy, attitude, knowledge, and social support for PA), PA behaviors (level, time, and practice of PA), and attributes (motor behavior, locomotor ability, physical fitness, endurance, flexibility, and anthropometry) [3,6].

Children (up to 12 years old) and adolescents (12 to 18 years old) represent two populations with particular characteristics [2,11,16]; thus, we synthesized the data according to age groups. The relevance of mapping different types of intervention strategies is based on age groups [13]. The strategies were organized in physical education classes; other strategies focused on changing the school environment, used electronic media, extracurricular PA sessions, and extracurricular sessions of health education, and focused on family and/or community [17]. Similarly, studies were organized by the outcome groups of PL dimensions, aiming at summarizing studies that evaluated and reported the effect of the intervention for each PA and PL outcome.

## 3. Results

After the electronic search, we found 3713 potential titles and abstracts (2944 studies after exclusion of duplicates). At the end of the first evaluation stage, 97 studies were considered eligible, and full-text reading was performed. Forty-three of these studies were not considered eligible, mainly because they did not measure the outcomes of interest (14 studies) or were not intervention studies (11 studies). Nine references were added after reading the reference lists of the retrieved documents or website searches. Thus, 63 documents (54 articles and nine theses and dissertations) met the eligibility criteria, representing 42 different studies that were summarized in this review [18,19,20,21,22,23,24,25,26,27,28,29,30,31,32,33,34,35,36,37,38,39,40,41,42,43,44,45,46,47,48,49,50,51,52,53,54,55,56,57,58,59,60,61,62,63,64,65,66,67,68,69,70,71,72,73,74,75,76,77,78,79,80] (Figure 1).

Overall, 17 and 25 intervention studies were performed in children and adolescents, respectively. In the study design, most studies were randomized controlled trials (*n* = 22; 52.4%). The sample size ranged from 17 to 2447 individuals, with 45.2% of the included studies consisting of samples with fewer than 100 participants (*n* = 19) and schoolchildren aged under 12 years (64.7%). Most interventions were performed in schools in the southern (40.5%) region of Brazil. Only one intervention study (with four reports) was focused on girls exclusively [28,29,30,31,32], and all others considered girls and boys combined. Thirteen (31.0%) studies performed interventions based on theoretical models (e.g., socio-ecological theory or the health-promoting schools approach). A total of 30 (71.4%), 19 (45.2%), and 6 (14.3%) of the included studies measured the outcomes of PA attributes, PA behavior, and modifiable PA-related factors, respectively. No study has evaluated PL using specific instruments/protocols for its assessment (Table 1).

Twenty-six intervention strategies were identified and organized into six dimensions. Interventions related to physical education classes were conducted in both age groups, which involved recreational activities with games, toys, and scavenger hunts that stimulated body movements [27,50,69,77,80]. Children and adolescents were provided with health education sessions such as lectures, debates, discussions, and dynamics on healthy eating, physical activity, and health [18,19,23,24,25,26,27,28,29,30,31,32,33,41,42,51,54,55,61,67,69,72,73,74]. In addition, there was teacher training in both student age groups [18,22,34,35,36,37,38,39,40,43,44,45,46,47,48,49,51,60,78,79] (Table 2).

Warm-up, aerobic, isometric, dynamic, postural, strength, endurance, stretching, and relaxation exercises were more frequent in adolescents [23,24,25,26,28,29,30,31,32,33,37,38,39,40,52,54,55,62,63,64,65,67,70,71,72,73,74,75,79], totaling to 25 different studies. Interventions focused on playing activities (e.g., jumping, dancing, and cooperative games) were frequent in children [18,21,41,42,43,53,56,57,58,59,60,61,77,80], with 13 studies. Interventions in both children and adolescents resulted in changes in the school environment; however, with children, interventions were focused on sports materials (ropes, ball, bows, cone, mat, tape and rubber bands, and track designed with colors) [56,59,66,68,76]. Interventions with adolescents focused on access to booklets, pamphlets, posters, banners, and exercise guides [19,20,22,23,24,25,26,28,29,30,31,32,33,34,35,36,44,45,46,47,48,49,51,78]. Only adolescents had the opportunity to interact electronically with instructional SMS for PA practice with motivational messages about environmental changes and access to information on healthy eating habits and creation of an electronic diary of PA and food [20,28,29,30,31,32,33,44,45,46,47,48,49,67]. Some studies have explored aspects of children’s motor skills [66,76] and adolescents’ postures [74,75]. Community health workers made home visits for children [18]. Parents were also consulted by researchers on the health behavior of their children from the intervention programs [27] (Table 2).

Table 3 summarizes the PL investigated in the studies according to age group. The components most frequently evaluated in interventions were PA attributes [18,19,20,21,23,24,25,26,27,28,29,30,31,32,37,41,42,44,45,48,50,52,53,54,56,57,58,59,61,62,63,64,65,66,67,68,69,70,71,72,75,76] (30 studies; 71.4%). In general, children were evaluated in terms of fine and global motor skills, balance, body schema, and spatial and temporal organization [50,56,76,80]. Muscle and cardiorespiratory endurance have been investigated more in adolescents [23,24,25,26,44,45,48,52,53,54,70,71,72] than in children [27,57,66]. Flexibility, physical and cardiorespiratory fitness, agility, BMI, weight, and height were variables assessed in both age groups [18,19,20,23,24,25,26,27,28,29,30,31,32,33,41,42,62,64,65,66,67,69,72]. Most studies evaluated PL components and reported that the intervention had an effect on them [20,21,27,41,42,50,53,54,56,57,58,59,61,62,63,64,65,66,69,70,71,72,75,76] (Table 3).

Of the 42 studies, 19 evaluated PA behavior, with more studies involving adolescents [18,23,24,25,26,28,29,30,31,32,33,34,37,38,39,40,43,44,45,46,47,48,49,51,55,60,61,63,67,73,74] than children [18,43,60,61]. The total duration of PA and moderate-to-intense PA practices was observed in both populations. Active commuting to/from school [44,45,46,47,48,49,73,74] was evaluated only in adolescents. Despite the purpose of evaluating the behavioral component of PA, 13 studies did not report the effect of the intervention [18,43,44,45,46,47,48,49,51,60,61,73,74] (Table 3).

PA-related factors were evaluated with children in only two studies [60,77]; however, 12 studies with adolescents reported results on them [19,22,23,24,25,26,28,29,30,31,32,33,37,38,39,40,51,75]. The psychological domain related to body perception, self-esteem, self-efficacy, perception of the environment, and satisfaction with PA was evaluated; however, several studies did not report any effect of these outcomes in adolescents [19,22,23,24,25,26,28,29,30,31,32,33,40,51]. Studies have reported a significantly positive effect on self-perception of posture [75] and PA attitude [37,38,39]. Nevertheless, the social domain has been exclusively studied among adolescents [22,28,29,30,31,32,33,37,38,39,40], with a positive association of the support indicator of parents, friends, and teachers [37,38,39]. Finally, knowledge on PA was assessed in children and adolescents, with an effect reported only in the latter age group [34,35,36]. The positive effect of intervention on PA-related factors has been reported only in a few studies with adolescents [34,36,37,38,39,75] (Table 3).

## 4. Discussion

### 4.1. Summary of Evidence

In the present review, 42 different intervention studies were conducted with Brazilian school-aged children and adolescents over more than two decades, with the aim of promoting PA and/or other PL components (mainly PA attributes, such as physical fitness). However, none of them evaluated PL using a specific method (i.e., using specific instruments and protocols) that addressed all PL components or different parts of them. Studies on PL have increased in high-income countries over the last few years, particularly with measurement and intervention models to promote the relevant components of individual and population health status [81]. However, our results reinforce the results of previous reviews that highlighted insufficient evidence on interventions for PA and/or PL in children and adolescents from low- and middle-income countries, such as Brazil [10,11]. Considering that the WHO highlights that PL is an objective of the global PA action plan [1], studies on this topic are urgent, particularly on the literature gaps highlighted in this review.

Most of the included interventions focused on PA attributes, such as physical fitness [18,19,20,21,23,24,25,26,27,28,29,30,31,32,33,41,42,44,45,48,49,50,52,53,56,57,58,59,61,62,63,64,65,66,67,68,69,70,71,72,75,76]. PA attributes related to motor competence were components studied only in children, such as motor/locomotor skills, body scheme, and balance; cardiorespiratory fitness, resistance, flexibility, and anthropometric indicators were components evaluated mainly in adolescents. Interventions should consider the physiological development and psychomotricity principles of each phase of life [2,16] because PA attributes are agents that offer physical and motor components for regular and diverse practices in PA [6]. However, Whitehead [2] argues that components of the psychological, social, and cognitive domains are fundamental to promoting individuals’ PL. Thus, it is important to investigate PA attributes at all stages of life in combination with PA practice and other PL components to deepen the psychosocial and behavioral aspects of PL [15].

The present review found that only six of the 42 interventions analyzed components of the dimension of PA-related factors [19,22,23,24,25,26,28,29,30,31,32,33,37,38,39,40,44,45,46,47,48,49,60,75]. In particular, PA-related factors and behaviors have been insufficiently studied in populations under 12 years of age. The literature recognizes that measuring exposures or outcomes in children can be challenging [82,83]. Assessing the PL has a pertinent health value; however, it is an aspect that faces challenges, making it necessary to take a delicate look at this broad field, particularly on the socio-environmental aspects that are as important as physical characteristics that facilitate the inclusion of individuals in the context of body movement and, consequently, in PA [6,15,84]. Thus, researchers have revealed the need to enhance tools and approaches that improve the measurement of PL, as tools have been created and improved to cover the entire PL without the detriment of any component [7,85]. The health of younger individuals is a wide field to be explored, and this review points to this gap to be improved.

Few studies have reported success in significantly promoting PA-related factors in adolescents; these components were less investigated, and of the existing studies, few reported an effect. One of the studies that addressed these topics was the “*Fortaleza sua Saúde*” program, a multicomponent intervention based on the Health Promoting Schools approach that found positive effects in attitude toward PA and parental, friends, and teachers’ support for PA [37,38,39]. Although a systematic review of interventions from high-income countries found positive effects in modifiable PA factors in the young population [8], our results suggest that intervention strategies in Brazil need to be improved to achieve positive effects on these PL components. These components are essential for promoting active individuals and societies [1].

Changes in physical education classes were the main strategies implemented in interventions [23,24,25,26,27,28,29,30,31,32,33,37,38,39,40,50,52,54,55,60,62,63,64,65,69,70,71,72,73], followed by extracurricular exercise sessions [18,19,21,41,42,43,53,56,57,58,59,61,66,67,74,75,76]. Both strategies have been recognized as effective strategies to promote physical components of PL, such as physical fitness and motor behavior [86,87]. Other studies focused on extracurricular health education sessions [18,19,23,27,33,41,42,51,54,55,61,67,69,72,73,74] and teachers’ training on PA and health content in the curriculum [18,22,34,35,36,37,38,39,40,43,44,45,46,47,48,49,51,60], which were relevant strategies to address knowledge and attitudes on PA and health. We also highlighted the number of studies that focused on social and environmental changes in schools [19,22,23,24,25,26,28,29,30,31,32,33,34,35,36,37,38,39,40,44,45,46,47,48,49,51,54,56,59,66,68,76]. This is important because for a school to be a health-promoting environment, the school environment must promote the dissemination of information through its inclusion into the school curriculum, including knowledge and experiences that enable autonomy, well-being, and health [88].

Only three Brazilian interventions have focused on strategies that involve family members [20,27,28,29,30,31,32,33]. Although the importance of the family through the construction and consolidation of PL components, particularly in children, strategies that involve the family and school context seem to be a challenge at a large scale, mainly because the difficulties involve parents and the family in the intervention process and behavioral change [89,90].

### 4.2. Practical and Research Implications of the Review

To the best of our knowledge, this is the first review to synthesize information about the various components of PL. Thus, an agenda of research priorities for interventions in Brazilian schools can be identified based on the findings of this review:Interventions with strategies that focus on the principles and components of PL (i.e., motivation, confidence, physical competence, knowledge, and understanding to maintain PA throughout life) should be planned and implemented, which may help understand which strategies should be implemented to promote PL components and how they interfere with PA and health outcomes. This helps build evidence on the direct and indirect effects of PL on PA and health at the population level [6].Studies should prioritize the use of measurement protocols and instruments that broadly examine PL components because this review showed that no intervention simultaneously addressed behavior, associated factors, and attributes.Studies should be planned to evaluate the intervention effectiveness of modifiable PA factors, considering the psychological, socio-environmental, and cognitive domains of PA. The current literature provides limited evidence on this topic.Studies with school-aged children under 12 years of age are stimulated, which propose intervention strategies and the evaluation outcomes related to PA behaviors and modifiable PA factors, using reliable and validated instruments and techniques (e.g., direct observation of accelerometer-measured PA).Studies with gender-specific results on the implementation and effectiveness are relevant to understand whether the intervention process and results are different between boys and girls.Several studies have measured the intervention effect on PA behaviors or attributes (particularly physical fitness and BMI). Thus, a synthesis that evaluates the evidence quality (i.e., risk of bias and evidence level based on a systematic review approach) may be conducted, which is important for defining the effectiveness of school-based interventions on these outcomes.Studies that focus on the feasibility and implementation process of the different types of school-based intervention strategies of PA and PL components are stimulated, including strategies that are frequently used (e.g., extracurricular exercise sessions or teachers’ training) and those that are innovative (e.g., electronic media interventions). This may be used by politicians, professionals, managers, and society to understand the barriers and facilitators of the different strategies, helping them adapt according to individual, social, and contextual features.

## 5. Conclusions

This review found 63 documents related to 42 different intervention studies conducted with Brazilian schoolchildren. However, there are no intervention studies that cover all PL components in the same study. The strategies most used in the interventions were changes in physical education classes, health education sessions, and extracurricular PA sessions. Adolescents were observed with a greater number of interventions. The most studied PL was related to PA attributes, in contrast to the factors associated with PA in its psychological, socio-environmental, and cognitive domains, which were the least investigated components. This review highlighted gaps in the literature and the need to conduct interventions with strategies that target all components of PL, representing important elements for a research agenda that underlies school interventions that contribute to an active lifestyle.

## Figures and Tables

**Figure 1 ijerph-18-08349-f001:**
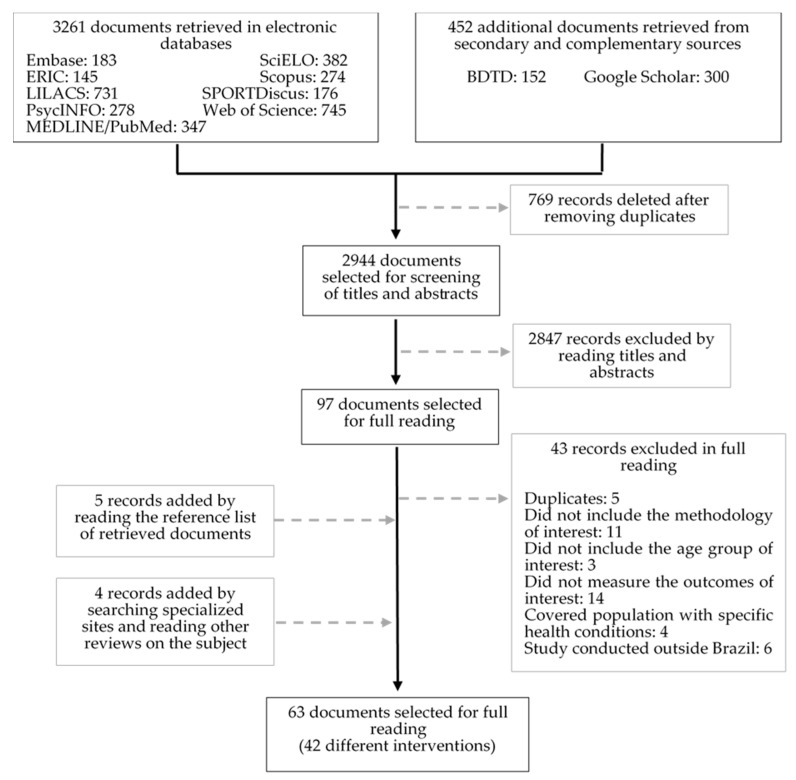
Flowchart of the selection process of studies (PRISMA-ScR). BDTD = *Biblioteca Digital Brasileira de Teses e Dissertações* (Brazil’s Digital Library of Thesis and Dissertations).

**Table 1 ijerph-18-08349-t001:** Publications and methodological characteristics of included studies according to age group.

Variables	All Studies (% of 42)	Studies in Children <12 Years (% of 17)	Studies in Adolescents ≥12 Years (% of 25)
**Year of Study**			
2000–2005	3 (7.1)	1 (5.9)	2 (8.0)
2006–2010	10 (23.8)	2 (11.8)	8 (32.0)
2011–2015	12 (28.6)	3 (17.6)	9 (36.0)
2016–June 2021	5 (11.9)	3 (17.6)	2 (8.0)
NA	12 (28.6)	8 (47.1)	4 (16.0)
**Brazil’s Region**			
North	3 (7.1)	0 (0.0)	3 (12.0)
Northeast	6 (14.3)	2 (11.7)	4 (16.0)
Midwest	1 (2.4)	1 (5.9)	0 (0.0)
Southeast	15 (35.7)	7 (41.2)	8 (32.0)
South	17 (40.5)	7 (41.2)	10 (40.0)
**Type of Study**			
RCT	13 (31.0)	7 (41.2)	6 (24.0)
Cluster RCT	9 (21.4)	4 (23.5)	5 (20.0)
NRCT	16 (38.1)	4 (23.5)	12 (48.0)
Quasi-experimental (without control)	4 (9.5)	2 (11.8)	2 (8.0)
**Sample Size (*n*)**			
>1000	7 (16.7)	2 (11.8)	5 (20.0)
501–1000	4 (9.5)	1 (5.9)	3 (12.0)
100–500	12 (28.6)	3 (17.6)	9 (36.0)
<100	19 (45.2)	11 (64.7)	8 (32.0)
**School Year**			
Elementary school only	33 (78.6)	17 (100.0)	16 (64.0)
High school only	8 (19.0)	0 (0.0)	8 (32.0)
Elementary and high school	1 (2.4)	0 (0.0)	1 (4.0)
**Program-based Interventions**			
Yes	13 (31.0)	5 (29.4)	8 (32.0)
No	29 (69.0)	12 (70.6)	17 (68.0)
**Study Object**			
PA	27 (64.3)	12 (70.6)	15 (60.0)
PA and healthy eating	15 (35.7)	5 (29.4)	10 (40.0)
**Intervention Time**			
11–12 months	2 (4.7)	0 (0.0)	2 (8.0)
8–10 months	7 (16.7)	1 (5.9)	6 (24.0)
5–7 months	9 (21.4)	5 (29.4)	4 (16.0)
2–4 months	16 (38.1)	5 (29.4)	11 (44.0)
<2 months	7 (16.7)	5 (29.4)	2 (8.0)
NA	1 (2.4)	1 (5.9)	
**Components of the Intervention**			
PL (specific instruments/protocols)	0 (0.0)	0 (0.0)	0 (0.0)
PA behavior only	7 (16.7)	2 (11.7)	5 (20.0)
PA behavior and associated factors	4 (9.5)	0 (0.0)	4 (16.0)
PA behavior and attributes	8 (19.0)	3 (17.6)	5 (20.0)
PA-associated factors only	1 (2.4)	1 (5.9)	0 (0.0)
PA-associated factors and attributes	1 (2.4)	0 (0.0)	1 (4.0)
PA attributes only	21 (50.0)	11 (64.8)	10 (40.0)

PA, physical activity; PL, physical literacy; RCT, randomized controlled trial; NRCT, nonrandomized controlled trial; NA, not applicable.

**Table 2 ijerph-18-08349-t002:** Intervention strategies and dimensions according to age groups.

Intervention	Physical Education Classes	Changes in the School Environment	Use of Electronic Media as a Strategy	Extracurricular PA Sessions	Extracurricular Health Education Sessions	Actions Focused on Family/Community
Children (<12 years)	Playing activities that encourage body movements (*n* = 4) [27,50,77,80]; Stimulation of free active movements during physical education classes (*n* = 1) [60].	Creation of physical spaces for PA (*n* = 1) [66]; Provision of sport materials (e.g., balls, cones, and rubber bands) in the school (*n*= 3) [56,59,76].		Stretching exercises, resistance, sports initiation games, balance, fine motor skills, and global and laterality (*n* = 2) [69,79]; PA sessions with activities (e.g., running, jumping, and dancing) or opposition games (*n* = 13) [18,21,41,42,43,53,56,57,58,59,60,61,77,80].	Lectures and educational sessions on PA, health, and nutrition (*n* = 4) [27,41,42,61]; Debates, discussions, dynamics, and practical experiences on PA and health (*n* = 1) [18].	Faculty training on PA, health, and nutrition (*n* = 3) [18,43,60]; Home visit by community health agents (*n* = 1) [18]; Consultation with parents about children’s health behaviors (*n* = 1) [27].
Adolescents (≥12 years)	PA sessions with pranks and games (*n* = 1) [69]; Warm-up exercises, including aerobics, dynamic movements, stretching, and relaxation (*n* = 25) [23,24,25,26,28,29,30,31,32,33,37,38,39,40,52,54,55,62,63,64,65,70,71,72,73,79].	Creation of physical spaces for PA (*n* = 10) [37,38,39,40,44,45,46,47,48,49,79]; Provision of sporting materials for sport practices (*n* = 13) [22,37,38,39,40,44,45,46,47,48,49,54,68,78,79]; Availability of booklets, pamphlets, posters, banners, exercise guides, books, and handouts (*n* = 22) [19,22,23,24,25,26,28,29,30,31,32,33,34,35,36,44,45,46,47,48,49,51,78].	SMS with instructions and motivational messages on PA (*n* = 7) [28,29,30,31,32,33,67]; Electronic diary of PA and food (*n* = 1) [20]; Dissemination of educational information about health on an electronic website (*n* = 7) [20,44,45,46,47,48,49].	Recreational PA with hitting, aerobics, and relaxation (*n* = 2) [19,67]; Posture education program (*n* = 2) [74,75].	Multiprofessional approach to nutrition and health education (*n* = 2) [19,67]; Lectures, workshops, and educational sessions on PA and health (*n* = 11) [23,24,25,26,28,29,30,31,32,33,69]; Debates, discussions, dynamics, and practical experiences on behavior in PA and health (*n* = 6) [51,54,55,72,73,74].	Training for teachers on PA and health (*n* = 15) [22,34,35,36,37,38,39,40,44,45,46,47,48,49,51,78,79]; Provision of informational materials for parents and teachers (*n* = 7) [20,28,29,30,31,32,33].

PA, physical activity; PL, physical literacy.

**Table 3 ijerph-18-08349-t003:** Components of PL investigated in the studies according to the age group.

Components of PL	Variables	Children (<12 years)		Adolescents (≥12 years)	
		Studies that Evaluated the Component	Studies that Reported the Effect of the Intervention	Studies that Evaluated the Component	Studies that Reported the Effect of the Intervention
PA-Associated Factors					
Psychological Domain	Posture self-perception			[75]	[75]
	Body perception			[19]	
	Self-esteem	[77]	[77]	[19,23,24,25,26]	
	Self-efficacy for PA			[22,23,24,25,26,28,29,30,31,32,33,37,38,39,40,78,79]	
	Perception of the environment for PA			[22,23,24,25,26,28,29,30,31,32,33,37,38,39,40,78,79]	
	Satisfaction for PA			[22]	
	Attitude for PA			[23,24,25,26,37,38,39,40,79]	[37,38,39]
Socio-Environmental Domain	Parental support for PA			[22,28,29,30,31,32,33,37,38,39,40]	[37,38,39]
	Support from friends for PA			[22,37,38,39,40]	[37,38,39]
	Teacher support for PA			[37,38,39,40]	[37,38,39]
Cognitive Domain	PA knowledge	[60]		[22,34,35,36,51]	[34,35,36]
	Physical attitude knowledge	[60]			
PA	PA level (active, insufficiently active, or inactive)	[18]	[18]	[51,55,63]	[51]
	PA practice	[43,60,61]	[43,60,61]	[23,24,25,26,28,29,30,31,32,33,34,35,36,37,38,39,40,44,45,46,47,48,49,63,67,73,74]	[44,45,46,47,48,49,73,74]
	Time spent in PA			[44,45,46,47,48,49,73,74]	[74]
PA Attributes	Fine motors skills	[50,56,76,80]	[50,56,76,80]		
	Global motor skills	[50,56,76,80]	[50,56,76,80]		
	Locomotor skills	[53]	[53]	[68]	
	Body scheme	[50,56,76]	[76]		
	Spatial organization	[50,76]	[76]		
	Temporal organization	[50,76]	[76]		
	General physical fitness	[21,27,59,61]	[59,61]	[63,65]	[63,65]
	Cardiorespiratory fitness	[66]	[66]	[23,24,25,26,54,71,72]	[54,71,72]
	Balance	[50,56,76]	[50,56,76]		
	Muscle endurance	[27,57,66]	[27,57,66]	[44,45,48,52,53,70,71,72]	[52,53,70,71,72]
	Flexibility	[57,58,66]	[57,58,66]	[54,70,71,72]	[54,70,71,72]
	Agility	[21]	[21]	[71]	[71]
	Posture			[75]	[75]
	Velocity	[21]	[21]		
	BMI	[18,27,41,42,66]	[27,41,42]	[19,20,23,24,25,26,28,29,30,31,32,33,62,64,65,67,69,72]	[20,62,64,69]
	Weight	[18,27,41,42,66]	[27,41,42]	[19,20,23,24,25,26,28,29,30,31,32,33,62,64,65,67,69,72]	[20,62,64,69]
	Height	[18,27,41,42,66]	[27,41,42]	[19,20,23,24,25,26,28,29,30,31,32,33,62,64,65,67,69,72]	[20,62,64,69]

PL, physical literacy; PA, physical activity; BMI, body mass index. The components of PL were operationalized in modifiable PA-related factors (e.g., motivation, self-efficacy, interpersonal support, and environmental factors), PA behavior (e.g., the weekly volume of PA), and PA attributes (e.g., physical fitness, motor behavior, and between others) [3,6].

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
