# Peer review of "Scoping Review on Interventions for Physical Activity and Physical Literacy Components in Brazilian School-Aged Children and Adolescents"

_ijerph, 2021, doi:10.3390/ijerph18168349_

Round 1
Reviewer 1 Report
The article is very good written. I guess it might be useful for all those who are interested at PA among children and adolescents. The only thing is, the text is based only/mostly on the Brazilian young population. I could see here the chances to show examples of other counties. In that way Authors could show the differences in some factors (e.g., cultural, policy, economic) and can discuss it in a wider manner. So I suggest to add some other studies (especially in the Introduction and Discussion sections) to fulfill the matter.
L. 3 - In the title, please specify the term students by adding "school".
L.38 - PA in not only "practice in sport" it is wider term meaning other forms of physical activities (e.g. leisure time physical activities; LTPA) undertaken also with family.
L. 41 - Not only "local culture" but also: strategy, region's policy and scope etc.
L. 47 - How do you understand "low-/middle-income countries"? And Here are some references' support would be well seen.
L. 63 - What does 'A' stand for? Please, give a note in the text.
L. 99 - search does not have a plural form.
L. 101 - Please, note there first appears 'VC' then in a line 106 'AC". Is it error or there are different initials for two people?
In the Figure 1 - What Authors mean by "searching specialized sites? Pleas, make it clear.
L. 158-159 - "Twenty-seven of them were not considered because they did not measure the outcomes of interest (n = 12) or were not
intervention studies (n = 7)". Is that ok? 12+7=19. Or there is other thought and I do not understand it?
L. 168 - give few examples of PA-related modifiable factors in the brackets.
L. 175 - you used twice the word "games" there.
L. 182 - from my point of view the word 'play' is more suitable in this sentence. Authors may use term “play” in the sense meaning a form of physical, playful activity, and a very first variety of game and sport. It refers to a range of spontaneous, voluntary, frivolous and non-serious activity. Some plays exhibits no goals or rules and are considered to be “unstructured” in the literature. For more information on play see the book of C. Garvey, (1999) Play, published by Cambridge, MA: Harvard University Press.
Literature suggested choosing to fulfill the text:
- Bronikowski & Bronikowska, Will they stay fit and healthy? A three-year follow-up evaluation of a physical activity and health intervention in Polish youth, Scandinavian Journal of Public Health, 2011; 39: 704–713.
- Bronikowski, et al. COMPARATIVE STUDY ON SELF-ASSESSMENT OF TEACHING COMPETENCIES OF PE STUDENT TEACHERS FROM POLAND AND KOSOVO, Journal Education, PE and Sport, 3(90), 2013,
- Cabak A, Woynarowska B. Physical activity in youth aged
11-15 in Poland and other European countries after 2002.
Phys Educ Sport 2004;4:335–60. - Sluijs van EM, McMinn AM, Griffin SJ. Effectiveness of
interventions to promote physical activity in children and
adolescents: systematic review of controlled trials. BMJ
2007;335(7622):703. - Sallis JF, McKenzie TL, Alcaraz JE, Kolody B, Faucette N,
HovellMF. The effects of a 2-year physical education program
(SPARK) on physical activity and fitness in elementary school
students. Am J Public Health 1997;87(8):1328–34. - Pluta et al. Associations between adolescents’ physical activity behavior and their perceptions of parental, peer and teacher support, Archives of Public Health (2020) 78:106 https://doi.org/10.1186/s13690-020-00490-3.
After minor changes I strongly recommend this article to be published.
Author Response
Dear reviewer of the IJERPH,
We thank the reviewers for their comments on the manuscript and the editor for the opportunity to address the comments and revising the manuscript. The comments have substantially contributed to the improvement of this version of the manuscript.
We hope this new version can address the high standard of the journal and we are looking forward to hearing your positive response. All modifications made in the text are highlighted in blue color.
Regards,
The Authors

Reviewer 2 Report
-
General comments:
This paper described the need to conduct interventions with strategies that target all components of PL, representing important elements for a research agenda that underlies school interventions that contribute to an active lifestyle. However, in relation to the contribution of the study to the literature, I did not get a sense from the article that the findings revealed anything other than what we already know. Please clarified.
Major comments:
- Why is it divided into groups (12 years)? In order to classify age, it should contain the content that "age affects physical activity pattern". Also, I think that gender is more important among the considerations for physical activity interventions. Wasn't there anything about gender in the literature?
- Were there any health behavior theoretical models/behavioral strategy studies that could be used to promote PA or other PL components?
- In addition, what is the theoretical model which supported the present study.
- The discussion part needs to be rewritten.
- The results and conclusions to achieve the purpose of this study are lacking.
Author Response

(The authors gave the same response as above.)

Reviewer 3 Report
INTRODUCTION
In general, I do not think the introduction does a great job of setting up the need for this review or the gap in evidence. I have a few specific points about how I think it can be improved – mostly regarding the need to clarify the concepts and relationship between physical literacy and physical activity. There are several recent papers and other reviews that can be reviewed and referenced throughout the paper.
First paragraph (line 32 -37) – I would consider softening the language or at least acknowledging the limited evidence base directly linking physical literacy, physical activity, and health. Although there is a theoretical and predicted link between these concepts, there is not much direct evidence that physical literacy is a determinant of health or physical activity.
Second paragraph (line 38-42) – measuring physical literacy involves more than measuring physical activity. Further explaining the potential connection with physical activity and the holistic nature of physical literacy would be beneficial here as physical literacy can be connected to other activities and quality of life outcomes beyond the ability to complete physical activity skills.
Third paragraph (lines 45-47) – I am not sure what the authors are trying to say here. What were the findings of the previous review? The “its” in the sentence “its effects on PA factors” is not really clear, physical literacy? Explaining the link and providing evidence (or lack of) on the relationship between physical activity and physical literacy would help here as well.
Line 47-48, why is it important to do research in low and middle-income countries specifically?
METHODS
First paragraph – was a protocol created or registered for this review?
Line 63-64 – what study was not conducted?
Line 66-67 – I would move the research question to the end of the introduction to more clearly establish the aims of this review.
Lines 69-78 – it is not clear what the outcome of interest was? Was the goal to describe the interventions, or describe the effectiveness of interventions to improve PA/PL?
Lines 87-91 (section 2.3) – how was the grey literature search conducted? What was searched, when, and how? The grey literature search is mentioned in the abstract but not in the manuscript.
Lines 98 – 104 (section 2.5) – consider providing a list of inclusion/exclusion criteria (or a table). The selection criteria are not clear as defined in the methods section.
RESULTS
Section 3 (PRISMA flow diagram) – what does the abbreviation BDTD mean? Are the articles indicated in the right-hand top box from grey literature sources?
Table 2 – some formatting issues and the columns are somewhat hard to read with odd spacing.
DISCUSSION
I suggest adding some general statements in the first paragraph of the discussion to summarize the aims of the review and the key findings. I think the discussion may also benefit from some sub-headings as it is hard to follow the different concepts and there is often limited flow between paragraphs.
Line 224 – what is “this outcome”, physical literacy?
Lines 231 – 234 – I am not really sure what the authors are trying to say here. How does this relate to physical literacy? Or is it that PA interventions improve motor competence (e.g., one component of PL).
Line 235-238 – I would combine these 2 sentences with the next paragraph.
Lines 239 – 244 – I think you need to be more explicit here about what domains of physical literacy have or haven’t been studied or reported in the literature, why they are important, and clearly identify what gaps remain in our understanding of the relationship between physical activity, physical literacy, and health. There are some recent reviews and papers that can be referenced.
Line 271 – why is it essential that principles of PL are used to define intervention strategies? What evidence is there to support this suggestion. This is similar to line 276, there is no reference or evidence to suggest a link between PL and health.
Author Response

(The authors gave the same response as above.)

Round 2
Reviewer 3 Report
There are still several issues with English language and style and the manuscript is challenging to understand in some places. Although the authors indicate revisions have been made based on my initial comments, there are several places where there have been no changes to the manuscript.
Line 75-76 - I am not sure what the authors mean by "this study was not conducted"
Line 79-80 (concept) - is not clear and requires revision
Line 97 - was the updated search done in the time since the first review of this manuscript?
The grey literature search strategy is still not provided.
Line 107 - what equation?
Line 240 - what is the basis for the statement that studies on PL in children in low/middle income countries is insufficient?
Author Response
Dear reviewer,
We thank you for their comments on the manuscript and the editor for the opportunity to address the comments and revising the manuscript. Our comments according to the reviewer’s suggestion are in the attached file.
We hope this new version can address the high standard of the journal and we are looking forward to hearing your positive response.
Regards,
The Authors
